# Enhanced Photocurrent of the Ag Interfaced Topological Insulator Bi_2_Se_3_ under UV- and Visible-Light Radiations

**DOI:** 10.3390/nano11123353

**Published:** 2021-12-10

**Authors:** Chih-Chiang Wang, Pao-Tai Lin, Fuh-Sheng Shieu, Han-Chang Shih

**Affiliations:** 1Department of Materials Science and Engineering, National Chung Hsing University, Taichung 40227, Taiwan; wilbur0913@gmail.com; 2International Agriculture Center, National Chung Hsing University, Taichung 40227, Taiwan; 3Department of Electrical and Computer Engineering, Texas A&M University, College Station, TX 77843, USA; paolin@ece.tamu.edu; 4Department of Chemical Engineering and Materials Science, Chinese Culture University, Taipei 11114, Taiwan

**Keywords:** Bi_2_Se_3_, Ag, surface plasmon resonance, photocurrent

## Abstract

Bi_2_Se_3_ is a topological quantum material that is used in photodetectors, owing to its narrow bandgap, conductive surface, and insulating bulk. In this work, Ag@Bi_2_Se_3_ nanoplatelets were synthesized on Al_2_O_3_(100) substrates in a two-step process of thermal evaporation and magnetron sputtering. X-ray diffractometer (XRD), high-resolution transmission electron microscopy (HRTEM), Raman spectroscopy, and x-ray photoelectron spectroscopy (XPS) revealed that all samples had the typical rhombohedral Bi_2_Se_3_. Field-emission scanning electron microscopy (FESEM)-energy dispersive x-ray spectroscopy (EDS), XPS, and HRTEM confirmed the presence of the precipitated Ag. The optical absorptance of Bi_2_Se_3_ nanoplatelets in UV-visible range decreased with the Ag contents. Results of photocurrent measurements under zero-bias conditions revealed that the deposited Ag affected photosensitivity. A total of 7.1 at.% Ag was associated with approximately 4.25 and 4.57 times higher photocurrents under UV and visible light, respectively, than 0 at.% Ag. The photocurrent in Bi_2_Se_3_ at 7.1 at.% Ag under visible light was 1.72-folds of that under UV light. This enhanced photocurrent is attributable to the narrow bandgap (~0.35 eV) of Bi_2_Se_3_ nanoplatelets, the Schottky field at the interface between Ag and Bi_2_Se_3_, the surface plasmon resonance that is caused by Ag, and the highly conductive surface that is formed from Ag and Bi_2_Se_3_. This work suggests that the appropriate Ag deposition enhances the photocurrent in, and increases the photosensitivity of, Bi_2_Se_3_ nanoplatelets under UV and visible light.

## 1. Introduction

Bismuth selenide (Bi_2_Se_3_) is a material with a narrow bandgap (~0.35 eV), a rhombohedral crystal structure, a unique gapless surface state, and a bulk electronic bandgap [1,2,3,4]. The unique properties of Bi_2_Se_3_ arise from the time-reversal symmetry and the spin-orbit coupling, which result in high electronic conductivity at the surface (three-dimensions) and edges (two-dimensions) [5]. Bi_2_Se_3_ has the following favorable properties: (1) low power dissipation, (2) photon-like and spin-polarized electrons, and (3) the quantum spin Hall Effect [6,7,8]. Owing to its unique optoelectronic properties, Bi_2_Se_3_ can be used in photodetectors [9], spintronic devices [10], and topological superconductors [11,12]. Bulk Bi_2_Se_3_ exhibits n-type semiconducting properties owing to the Se vacancies, which supply free electrons that induce a high bulk conductivity and suppress its surface electronic transportation [13,14]. Nanosized Bi_2_Se_3_ crystals exhibit a weaker bulk effect and greater electronic transportation because of their high surface-to-volume ratio [15]. The methods for synthesizing nanosized Bi_2_Se_3_ include: (1) thermal chemical vapor deposition (CVD) process [16], (2) successive ionic layer adsorption and reaction (SILAR) [17], (3) chemical bath deposition (CBD) [18], (4) electrodeposition [19], and (5) the solvothermal method [20]. Photodetectors are useful in optical information communications, imaging detection, and biodetection owing to the transit between photon and electron [21,22,23]; to be effective, they must have a high photoelectric conversion efficiency. A photodetector should (1) exhibit photo-absorptance over a wide range of wavelengths, (2) high photosensitivity, (3) high free-carrier mobility, (4) high photon-electron conversion efficiency, (5) a low operating voltage, and (6) long-term stability [24,25]. Bi_2_Se_3_ satisfies all of the requirements of a potential photodetector owing to its unique surface state and excellent surface electronic transportation. Various methods for enhancing its photocurrent have been reported upon. Pamu et al. found that Au or Ag nanodisks enhance the photocurrent of photosystem I (PSI) as a result of the plasmonic interaction between metal and PSI [26]. Peng et al. found that a semiconductor quantum dot of InAs/GaAs increases the photocurrent in an *n-i* Schottky device as a result of the Coulomb interaction [27]. Yan et al. found that a textured silicon wafer and periodic Ag nanoarrays increase the photocurrent of the silicon solar cell owing to their textured morphology and plasmonic effect [28]. Chakraborty et al. reported that Au increases the photocurrent in the CdSe nanowires owing to the plasmonic effect and Schottky field at the interface between the metal and the semiconductor [29]. These reports suggest that the photocurrent can be enhanced by the plasmonic effect, the Schottky field, and surface morphology. The photocurrent in, and enhancements of, Bi_2_Se_3_ have been studied and reported [30,31,32]. Wang et al. applied external strain to Bi_2_Se_3_ nanowires and thereby increased the photocurrent therein [33]. Gupta et al. found that the Ag nanoparticle-decorated Bi_2_Se_3_ that was synthesized by a chemical solution method exhibited an enhanced photocurrent at a bias of −10 V [34]. Chae et al. found that Bi_2_Se_3_/graphene exhibited an enhanced photocurrent with a source-drain voltage of 1.5V [35]. Liao et al. found that reduced graphene oxide increased the photocurrent of Bi_2_Se_3_ nanosheets [36]. Based on the enhancing factors of the plasmonic effect and the interaction between metal and semiconductors, the thermal CVD process that involves a catalyst-free vapor-solid mechanism is used herein to synthesize Bi_2_Se_3_ nanoplatelets, on which Ag is deposited by magnetron sputtering. The enhancement of the photocurrent in the Bi_2_Se_3_ nanoplatelets under UV and visible light by Ag was studied.

## 2. Materials and Methods

### 2.1. Fabrication of Pristine Bi_2_Se_3_ and Ag@Bi_2_Se_3_ Nanoplatelets

Pristine Bi_2_Se_3_ nanoplatelets were fabricated on an Al_2_O_3_ (100) substrate (0.5 × 0.5 mm^2^) using a catalyst-free vapor-solid mechanism by thermal evaporation in a quartz tube furnace. A mixture of precursor powders of 0.1g bismuth (purity = 99%, 4.78 × 10^−4^ mole, Merck, Darmstadt, Germany) and 0.1g selenium (purity = 99%, 1.27 × 10^−3^ mole, Alfa Aesar, Ward Hill, MA, USA) were placed in an alumina boat, which was placed in the heating zone at the center of the quartz tube and heated to 600 °C at a rate of 25 °C/min under 1.5 × 10^−2^ Torr, in which conditions were maintained for 60 min. The Al_2_O_3_ (100) substrate was placed upstream in the quartz tube at about 140 °C, 21 cm away from the alumina boat. Thus, pristine Bi_2_Se_3_ nanoplatelets were grown on the Al_2_O_3_ (100) substrate. Then, the synthesized system was slowly cooled to room temperature after a 60 min deposition process. An Ag thin film was deposited on a sapphire substrate for 60 s (110V, 4 mA) at 1 × 10^−1^ mbar. It was then measured using a profilometer (Dektak XT, Bruker, Billerica, MA, USA) to estimate the Ag deposition rate. Then, Ag was deposited on the Bi_2_Se_3_ nanoplatelets at a working distance of 35 mm by magnetron sputtering (110 V, 4 mA) using a 2-inch Ag target at room temperature under 1 × 10^−1^ mbar. Ag was deposited for 10, 15, 20, and 25s yielding Ag10s@Bi_2_Se_3_, Ag15s@Bi_2_Se_3_, Ag20s@Bi_2_Se_3_, and Ag25s@Bi_2_Se_3_ nanoplatelets, respectively.

### 2.2. Characterization of Nanoplatelets

The crystal structures of the pristine Bi_2_Se_3_ and Ag@Bi_2_Se_3_ nanoplatelets were determined using XRD (λ = 0.154 nm, 30 A, 40 kV, Bruker D2 PHASER) at 2θ = 10–60°, and HRTEM (JEOL JEM-2010, Tokyo, Japan). XPS (PerkinElmer model PHI1600 system, Waltham, MA, USA) and Raman spectroscopy (3D Nanometer Scale Raman PL Microspectrometer, Tokyo Instruments, INC., Tokyo, Japan) with a semiconductor laser (λ = 488 nm) were used to record the chemical binding energies and vibration modes of the chemical bonds. The surface morphology and EDS spectra of the pristine Bi_2_Se_3_ and Ag@Bi_2_Se_3_ nanoplatelets were obtained using FESEM (ZEISS Ultra Plus, Carl Zeiss Microscopy GmbH, Oberkochen, Germany). The optical absorptance of UV and visible light was recorded using a UV-visible spectrometer (Hitachi U3900-H, Hitachi Ltd., Tokyo, Japan) that was equipped with an integrating sphere.

### 2.3. Photocurrent Measurements

A semiconductor I-V characterizing analyzer (Keysight B2901A Precision Source/Measure Unit 100 fA, Keysight Technologies, Santa Rosa, CA, USA) recorded the photocurrents at 0 V bias under UV- and visible-light in the ambient environment. The sources of incident light were 30 cm-long UV (8 W, λ = 365 nm) and visible light (8 W, λ = 380–780 nm) LED lamps at 20 cm from the sample. The photocurrents in all samples were recorded in a dark room to eliminate any effect from stray lights. Silver paste was used as an electrode that was connected to a current analyzer using copper wires. The photocurrent in each sample was measured six times; each time, the light was on for 10 s and off for 10 s.

## 3. Results

### 3.1. XPS Analysis

Figure 1 present the XPS spectra of the Bi 4f, Se 3d, and Ag 3d of the pristine and Ag25s@Bi_2_Se_3_ nanoplatelets. Figure 1a display the spectra of the Bi 4f orbit of the pristine Bi_2_Se_3_ and Ag25s@Bi_2_Se_3_ nanoplatelets, revealing four significant peaks. The peaks at 157.94 and 163.26 eV are associated with the Bi 4f^7/2^ and Bi 4f^5/2^ in Bi_2_Se_3_ [16]; those at 158.69 and 164.30 eV correspond to the binding energies of the Bi–O phase in Bi_2_O_3_ [37]. Figure 1b show the XPS of Bi 4f in the pristine Bi_2_Se_3_ and Ag25s@Bi_2_Se_3_ nanoplatelets, from which the peaks are in similar positions. Figure 1c and d present the Se 3d orbits of the pristine Bi_2_Se_3_ and Ag25s@Bi_2_Se_3_ nanoplatelets. Three main peaks at 53.5, 54.3, and 55.1 eV are observed. The first two peaks correspond to Se 3d^5/2^ and Se 3d^3/2^ in Bi_2_Se_3_ [38], and the last one corresponds to the metal Se [39]. The metallic Se traps the free electrons and suppresses the photocurrent that arises from them. A weak peak at 58.66 eV is observed in Figure 1d, arising from the Se–O bonds [40]. These XPS spectra confirm the formation of the Bi_2_Se_3_ structure. The peaks of Bi–O and Se–O are attributed to natural formations on the surface of the sample in an ambient environment. Figure 1e present the Ag 3d binding energy in Ag25s@Bi_2_Se_3_ nanoplatelets. Peaks at 367.55 and 373.58 eV with an energy separation of 6.03 eV correspond to the Ag 3d^5/2^ and Ag 3d^3/2^ of metallic Ag. The binding energies of Ag 3d^5/2^ and Ag 3d^3/2^ are lower than that of bulk Ag (~368 eV). The shift of the binding energy is relative to the electronegativity of the elements and is responsible for the change in the charge around the atom [41]. The larger electronegativity of the element, the blue-shift of the binding energy. Ag (1.93) has a lower electronegativity than Bi (2.02) and Se (2.55). Therefore, the binding energies of Ag 3d^5/2^ and Ag 3d^3/2^ shift toward lower energy (red-shift). Table 1 show the atomic percentages of Bi, Se, and Ag, confirming the presence of Ag; the portion of Ag increases with the deposition time.

### 3.2. Morphological Analysis

Figure 2a,b present the morphologies of the pristine Bi_2_Se_3_ and Ag25s@Bi_2_Se_3_ nanoplatelets that are obtained using FESEM. Both samples have a hexagonal structure which is typical of rhombohedral Bi_2_Se_3_. FESEM-EDS detects the presence of Ag along with the Bi and Se. The estimated thickness (40 nanoplatelets) and size (40 nanoplatelets) of each nanoplatelet of pristine Bi_2_Se_3_ is 6.9 and 8.3 nm; those of Ag25s@Bi_2_Se_3_ are 444.8 and 488.5nm, respectively. Bi_2_Se_3_ has the same layered structures as graphite. Each layer comprises five stacked atomic layers Se^1^–Bi–Se^2^–Bi–Se^1^ and is called the quintuple layer (QL). The thickness of each QL is 0.955 nm [42], indicating that pristine Bi_2_Se_3_ and Ag25s@Bi_2_Se_3_ nanoplatelets have 7.22 and 8.69 OLs, respectively. Figure 2c present the thickness of the Ag thin film that is deposited on a sapphire substrate at 0.44W for 60s (110 V, 4 mA) at 1 × 10^−1^ mbar. The ideal Ag thicknesses in pristine Bi_2_Se_3_, Ag10s, Ag15s, Ag20s, and Ag25s@Bi_2_Se_3_ nanoplatelets are 0, 3.3, 5, 6.7, and 8.3 nm, respectively. Therefore, the Ag was not easily observed in the FESEM images.

### 3.3. Analysis of Crystal Structures and Surface Morphologies

Figure 3 present the XRD patterns of the pristine Bi_2_Se_3_ and Ag@Bi_2_Se_3_ nanoplatelets. Nine significant peaks are observed at 2θ values of 18.55°, 25.03°, 27.91°, 29.39°, 40.28°, 43.07°, 43.77°, 47.67°, and 53.61°, corresponding to the planes (006), (101), (104), (015), (1010), (0111), (110), (0015), and (205) of the rhombohedral Bi_2_Se_3_ structure (JCPDS 89-2008). No significant peak of oxides or silver are observed because the content of oxides and Ag are less than the limit of detection (~5%) by XRD. The grain sizes of pristine Bi_2_Se_3_ and Ag@Bi_2_Se_3_ nanoplatelets are estimated at 14.56±7.9% nm using the Williamson-Hall equation [43] and plotted in the inset in Figure 3. Table 2 present the lattice constants a (= b) and c and the c/a ratios of the Bi_2_Se_3_ nanoplatelets with various Ag contents.

Figure 4a–c present the HRTEM images and selected area diffraction (SAD) patterns of the pristine Bi_2_Se_3_ nanoplatelets. Figure 4a present the hexagonal morphology that is typical of rhombohedral Bi_2_Se_3_. The d-spacings of 0.334 and 0.317 are estimated from Figure 4b, corresponding to the Bi_2_Se_3_ (012) and Bi_2_Se_3_ (009) planes. Figure 4c present the SAD patterns and reveals Bi_2_Se_3_ (101), Bi_2_Se_3_ (012), and Bi_2_Se_3_ (110) planes. Figure 4d–f present HRTEM images and SAD patterns of Ag25s@Bi_2_Se_3_ nanoplatelets. Figure 4d show that the Ag20s@Bi_2_Se_3_ nanoplatelets have the same morphology as the pristine ones, which is shown in Figure 4a. Figure 4e reveal the estimated d-spacings of 0.341 and 0.207 nm, which correspond to Bi_2_Se_3_ (012) and Ag (200), respectively, as shown in the blue-line region in Figure 4e. Figure 4f present the SAD pattern of Ag20s@Bi_2_Se_3_ nanoplatelets, which corresponds to the Bi_2_Se_3_ (012), Bi_2_Se_3_ (009), and Bi_2_Se_3_ (110) planes. The results of XRD and HRTEM reveal that the crystal structures of pristine Bi_2_Se_3_ and Ag20s@Bi_2_Se_3_ nanoplatelets are the typical rhombohedral Bi_2_Se_3_ structure. The Ag phase is also observed in Figure 4e.

### 3.4. Raman Analysis

Figure 5a,b present two typical Raman peaks of rhombohedral Bi_2_Se_3_ at $E_{g}^{2}$ and $A_{1g}^{2}$ from the pristine Bi_2_Se_3_ and Ag25s@Bi_2_Se_3_ nanoplatelets [16]. Bi_2_Se_3_ belongs to the space group ($R\bar{3}m/D_{3d}^{5}$). The atoms in a QL are bonded covalently [44]; Van der Waals forces bind the QLs [16]. The zone-center phonon formula of Bi_2_Se_3_ is $\chi =2E_{g}+2A_{1g}+2E_{u}+2A_{1u}$ [42], where $2A_{1g}$ and $2E_{g}$ are the Raman-active modes and $2A_{1u}$ and $2E_{u}$ are the infrared-active modes. The $\;A_{1g}^{2}$ mode is the out-of-plane stretching symmetry mode with the vibration of the Se^1^ and Bi^1^ atoms in opposite directions. Therefore, the $A_{1g}^{2}$ mode has a short displacement (Figure 5a). The $E_{g}^{2}$ mode is the in-plane bending symmetry mode shearing the upper Se^1^ and Bi^1^ atomic layers, which vibrate with a longer displacement than in the $A_{1g}^{2}$ mode and have a large binding energy (Figure 5a) [45]. These results suggest that the rhombohedral Bi_2_Se_3_ formed successfully, and the Ag has no significant effect on the binding structure.

### 3.5. Analysis of Optical Properties

Figure 6a present the absorptance spectrum of the pristine Bi_2_Se_3_ and Ag@Bi_2_Se_3_ nanoplatelets. The absorptance of the nanoplatelets decreases as the Ag deposition time increases. Appendix A reveal the absorptance of Ag/Bi_2_Se_3_ thin film as simulated in TFCalc software. The simulation involves (1) a Bi_2_Se_3_ thin film with a thickness of 10 nm, (2) Ag thin films with a thickness of 0, 2, 4, 6, 8, and 10 nm, and (3) a sapphire substrate. The absorptance of the Ag@Bi_2_Se_3_ thin films decreases as the Ag thickness increases, consistent with the experimental results (Figure 6a). This decrease in absorptance is attributed to the increase in reflectance by the decoration with Ag. The absorptance of the Ag@Bi_2_Se_3_ nanoplatelets is therefore suppressed. Figure 6b present the normalized absorptance at various Ag deposition times. The Ag@Bi_2_Se_3_ nanoplatelets have a broad band from 375 to 420 nm that is centered at ~398 nm.

In order to understand the effect of the Ag thin film on the optical absorptance of Bi_2_Se_3_, a simulation is performed, and its results are presented in Appendix A. The simulated structure is a Au thin film/Bi_2_Se_3_ thin film (10 nm)/sapphire substrate. The simulation involves (1) Ag thin films with a thickness of 0, 2, 4, 6, 8, and 10 nm, and (2) a sapphire substrate. A significant peak at 400 nm is observed while the Ag thickness increases, as shown in Appendix A, consistent with the experimental results (Figure 6b). The broad peak from 375 to 420 nm that is centered at ~398 nm (Figure 6b) is therefore confirmed to be caused by the Ag and attributable to the surface plasmon resonance [46].

### 3.6. Photocurrent under the UV and Visible Light

Figure 7a,b plot the measured photocurrents under UV (8 W, λ = 365 nm) and visible light (8 W, λ = 380–780 nm); the measurements are performed six times. Each measurements cycle lasts 20 s, comprising 10 s under illumination and 10 s without illumination. The substrate bias is 0 voltage during the measurement of photocurrent; therefore, the photocurrents are spontaneously generated by the incident light. The photocurrents in pristine Bi_2_Se_3_ under UV and visible light are lower than those in the Ag@Bi_2_Se_3_ nanoplatelets, as shown in Figure 7a,b. The photocurrent increases with Ag content from 0 to 7.1 at.%, and then decreases as the Ag content increases further to 8.2 at.%. Figure 7c plot the photocurrent as a function of Ag content. The highest photocurrents at 7.1 at.% Ag exceed those in the pristine nanoplatelets by factors about 4.25 and 4.57 under UV and visible light, respectively. Additionally, the photocurrents at 7.1 at.% Ag under visible light exceed those under UV light by a factor of about 1.7. These results reveal that a particular Ag content can increase the photocurrent in Bi_2_Se_3_ nanoplatelets under UV and visible light and that Ag@Bi_2_Se_3_ nanoplatelets have higher photosensitivity under visible light than that under UV light.

### 3.7. Mechanism of Photocurrent Enhancement

Bi_2_Se_3_ is a semiconductor with a narrow bandgap (~0.35 eV) [47], in which the photo-induced electron-hole pairs are readily generated by UV and visible light. A Schottky field can be formed at the interface between Ag and Bi_2_Se_3_, and it dissociates the photo-induced electrons and holes in Bi_2_Se_3_ [29]. This dissociation process generates free electrons and suppresses their recombination with holes. Surface plasmon resonance (SPR) (Figure 6b) is observed in the Ag@Bi_2_Se_3_ nanoplatelets. SPR is the collective oscillation of electrons in the conduction band after the absorption of light with a particular incident energy, generating unstable/free electrons. These free electrons can be smoothly transported at the conductive surface that contributed to the Ag and nanosized Bi_2_Se_3_ platelets [15], enhancing the photocurrent. This photocurrent enhancement has the following causes: (1) the narrow bandgap of the pristine Bi_2_Se_3_ nanoplatelets, (2) the Schottky field at the interface between Ag and Bi_2_Se_3_, (3) the SPR effect of the Ag decoration, and (4) the highly conductive surface contributing to the Ag and Bi_2_Se_3_ nanoplatelets. Figure 8 present the proposed transmission path.

Notably, the photocurrent under visible light exceeds that under UV. The absorptance of nanoplatelets under visible light exceeds that under UV (Figure 6a), indicating that visible light can be efficiently used by the nanoplatelets. The high-energy incident light (UV) generates high concentrations of free carriers close to the surface. These free carriers then recombine through surface states because of the high carrier density, reducing the total photocurrent [48]. A high Ag content (8.1 at.%) is associated with lower photocurrent under UV or visible light. The SPR enhances the near-field intensity and increases the concentrations of unstable/free electrons. Therefore, higher densities of electrons are generated in the surface region, and these electrons subsequently recombine with the holes, suppressing the photocurrent.

## 4. Conclusions

The Ag@Bi_2_Se_3_ nanoplatelets exhibit a significantly higher photocurrent under UV and visible light. The photocurrents of Ag@Bi_2_Se_3_ nanoplatelets with 7.1 at.% Ag under UV and visible light were approximately 4.25 and 4.57 times larger than those in pristine Bi_2_Se_3_ nanoplatelets. The resulting photocurrent in the Ag@Bi_2_Se_3_ nanoplatelets with 7.1 at.% Ag under visible light is 1.72-fold compared to that of pristine nanoplatelets under UV light. The presence of Ag and metallic Se is confirmed by XPS, FESEM, and HRTEM images. Free electrons are trapped by metallic Se, reducing the photocurrent. The rhombohedral Bi_2_Se_3_ crystal structure is confirmed by XRD, HRTEM, and Raman spectra. The Ag decoration has no effect on the Bi_2_Se_3_ crystal structure. The absorptance that is recorded by the UV-visible spectrometer reveals that increasing the Ag content reduces their total absorptance, affecting their light-to-electron transformation efficiency, suppressing the generation of the photocurrent. The SPR effect of Ag is evident in the absorptance spectra; it generates unstable/free electrons, enhancing the photocurrent in the nanoplatelets. The appropriate Ag content significantly enhances the photocurrent of pristine Bi_2_Se_3_ nanoplatelets because of the following: (1) the narrow bandgap of Bi_2_Se_3_ nanoplatelets, (2) the Schottky field at the interface between Ag and Bi_2_Se_3_, (3) the surface plasmon resonance that is caused by Ag, and (4) the highly conductive surface. The photocurrent was suppressed for the following reasons: (1) the high carrier densities that are generated by the high-energy incident light and SPR increase the rate of recombination between electrons and holes; (2) metallic Se may act as a recombination center; and (3) the absorptance of the Bi_2_Se_3_ nanoplatelets is reduced by Ag decoration. This work suggests that the incorporation of a particular amount of Ag enhances the photocurrent in Ag@Bi_2_Se_3_ nanoplatelets; this phenomenon may have immediate application in the detection of UV and visible light.

## Figures and Tables

**Figure 1 nanomaterials-11-03353-f001:**
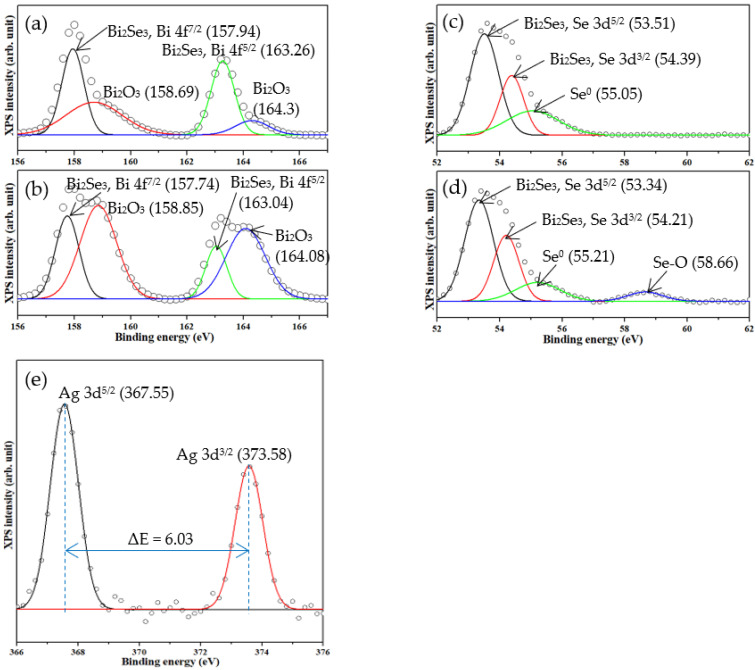
XPS spectra of Bi 4f, Se 3d of Bi_2_Se_3_ of (**a**,**c**), and of Ag25s@Bi_2_Se_3_ of (**b**,**d**); (**e**) Ag 3d of Ag25s@Bi_2_Se_3_ nanoplatelets.

**Figure 2 nanomaterials-11-03353-f002:**
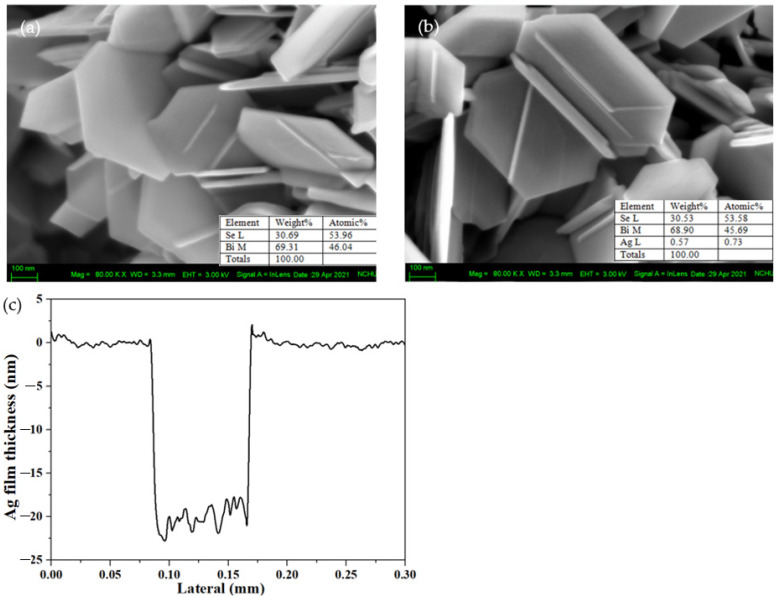
FESEM images of (**a**) Bi_2_Se_3_ and (**b**) Ag25s@Bi_2_Se_3_ nanoplatelets. Insets are the EDS results of Bi, Se, and Ag. (**c**) Thickness of Ag film deposits on the sapphire substrate.

**Figure 3 nanomaterials-11-03353-f003:**
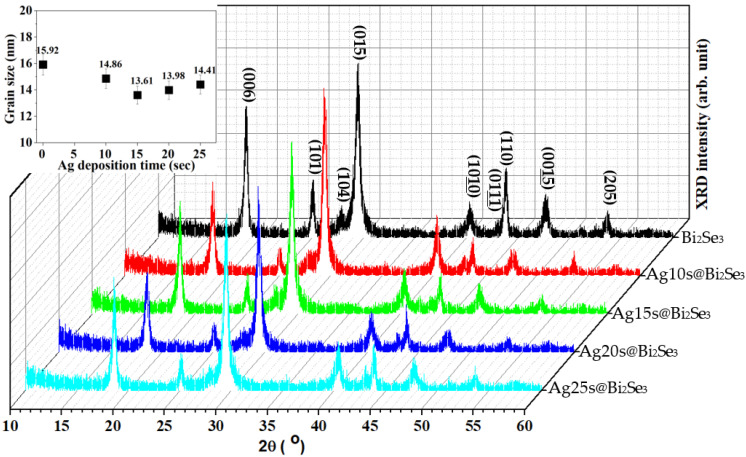
XRD patterns of Bi_2_Se_3_ and Ag@Bi_2_Se_3_ nanoplatelets. Inset are the grain size variations estimated by the W–H plot.

**Figure 4 nanomaterials-11-03353-f004:**
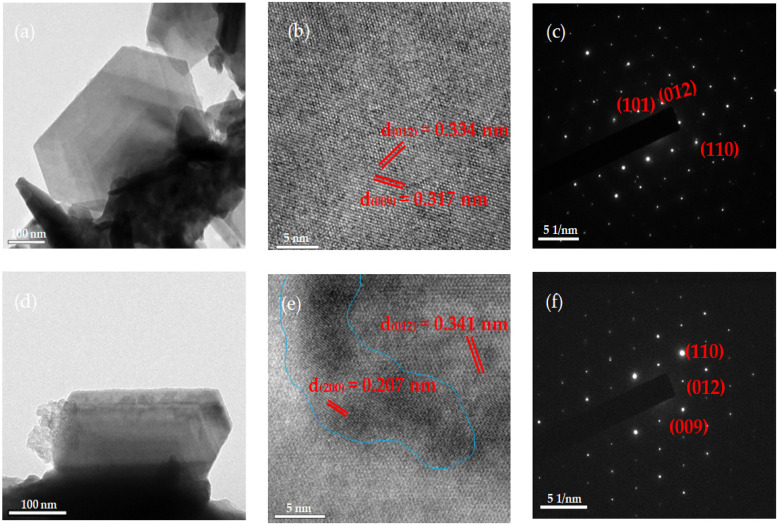
HRTEM images of (**a**,**b**) Bi_2_Se_3_ and (**d**,**e**) Ag20s@Bi_2_Se_3_; SAD patterns of (**c**) Bi_2_Se_3_ and (**f**) Ag20s@Bi_2_Se_3_ nanoplatelets.

**Figure 5 nanomaterials-11-03353-f005:**
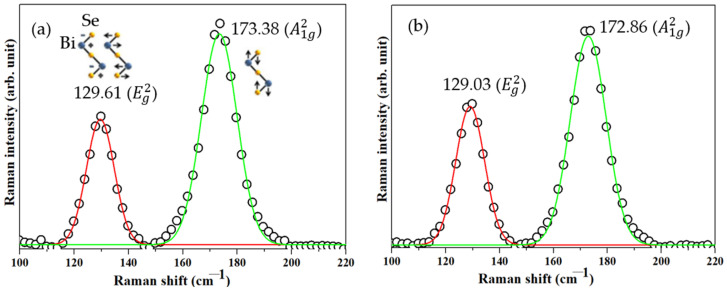
Raman spectra of (**a**) Bi_2_Se_3_ and (**b**) Ag25s@Bi_2_Se_3_ nanoplatelets.

**Figure 6 nanomaterials-11-03353-f006:**
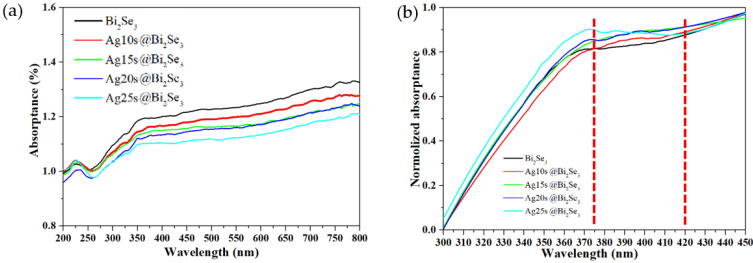
(**a**) UV-visible absorptance and (**b**) normalization of absorptance of pristine Bi_2_Se_3_ and Ag@Bi_2_Se_3_ nanoplatelets.

**Figure 7 nanomaterials-11-03353-f007:**
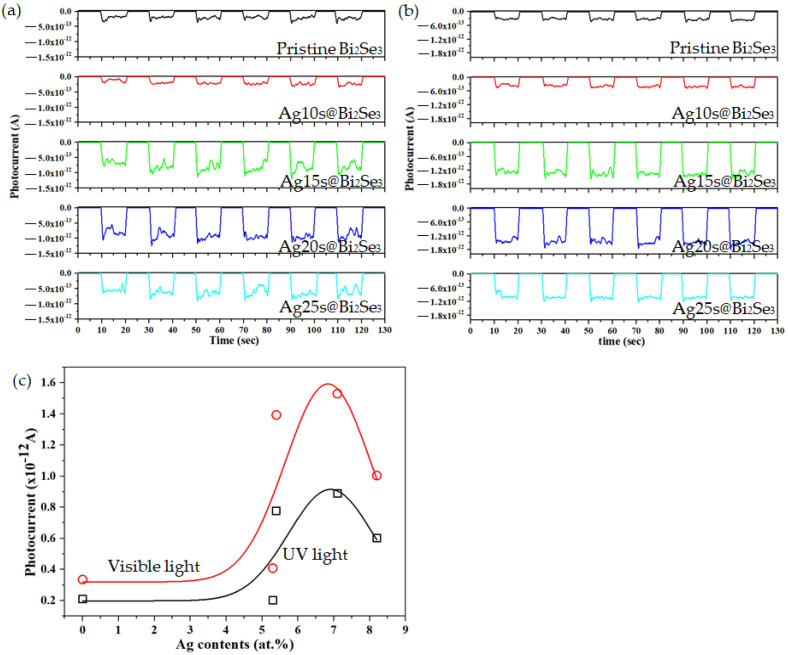
Photocurrents versus measured time under (**a**) UV, (**b**) visible light and (**c**) The variations of the photocurrent versus Ag contents (at.%).

**Figure 8 nanomaterials-11-03353-f008:**
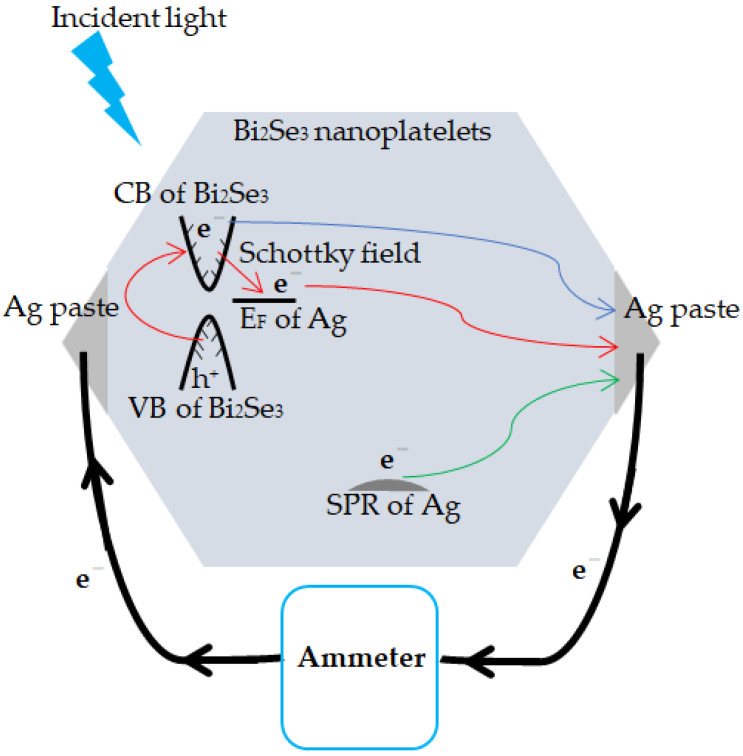
The proposed transmission path of electrons in the Ag@Bi_2_Se_3_ nanoplatelets.

**Table 1 nanomaterials-11-03353-t001:** Atomic % of pristine Bi_2_Se_3_ and Ag@Bi_2_Se_3_ nanoplatelets of Bi, Se, and Ag.

Sample	Bi (at.%)	Se (at.%)	Ag (at.%)
Bi_2_Se_3_	47.1	52.9	0
Ag10s@Bi_2_Se_3_	52.3	42.3	5.3
Ag15s@Bi_2_Se_3_	53.5	41.1	5.4
Ag20s@Bi_2_Se_3_	53.5	39.4	7.1
Ag25s@Bi_2_Se_3_	52.6	39.2	8.2

**Table 2 nanomaterials-11-03353-t002:** Lattice constants of a (= b) and c, and c/a ratios of Bi_2_Se_3_ nanoplatelets with various Ag contents.

Ag Contents (at.%)	a (= b) (nm)	c (nm)	c/a
0	0.4133	2.8675	6.9381
5.3	0.4129	2.8736	6.9595
5.4	0.4128	2.8721	6.9576
7.1	0.4129	2.8736	6.9595
8.2	0.4131	2.8705	6.9486

## Data Availability

Data is contained within the article.

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
