# Peer review of "Enhanced Photocurrent of the Ag Interfaced Topological Insulator Bi2Se3 under UV- and Visible-Light Radiations"

_nanomaterials, 2021, doi:10.3390/nano11123353_

Round 1

Reviewer 1 Report

Re: Nanomaterials - 1476922

         Enhanced photocurrent of the Ag interfaced topological insulator Bi2Se3 under UV- and visible-light radiations

            In this paper Wang et al., synthesize Ag/ Bi2Se3 nanoplatelets and they characterize their samples using several techniques such as XRD, HRTEM, Raman spectroscopy, and XPS. After a detailed and analytical characterization, they performed electrical measurements in order to record the photocurrent under zero-bias and study the effect of UV and visible light. The main result is the observation of an enhanced photocurrent for 7% Ag for both UV and visible. The authors discuss in detail the reasons for the enhancement and propose a mechanism for the transmission of the electrons. It is a very detailed and systematic work. The manuscript is clearly written and the results are very well argued. Therefore, I recommend it for publication in Nanomaterials.

Reviewer 2 Report

This work has been focused on the preparation and characterization of Bi2Se3 and Ag-Bi2Se3 photocataysts for the enhanced photocurrent density for the photocatalytic property enhancement. This is possible by the interface junction between Bi2Se3/Ag. But authors should add more data on the photoexcited electron capturing ability of Ag due to its low energy level compared with Bi2Se3. This can be carried out by getting time-resolved photoluminecsence decay curve fitting. Geberally, Enlgish should be revsied by native speaker.

Reviewer 3 Report

The authors report enhanced photocurrents in Ag covered Bi2Se3 nanoplatelets subject to UV- and visible-light irradiation. This is presumed to be an effect due to surface plasmon resonance. The study presents a number of structural (XRD,HRTEM,Raman), morphological (FESEM-EDS) and chemical (XRD) characterizations in order to confirm the presence of Ag covering the  Bi2Se3 nanoplatelets. 

The results presented here showing an increased photosensitivity are of interest for the design of efficient photodetectors. It may be considered for publication, pending some issues are resolved:

1. The authors mention the bandgap of 0.35 eV for Bi2Se3, a value taken from the literature. Can the authors provide a bandgap value for the present case ?

2. It seems that there is a very large variation of the photocurrent between Ag content at.% of 5.3 and 5.4. It is rather unfortunate to have the two values so close. However, why there is this sharp variation ? What kind of fit the authors used for these data values ? 

3. Some figures can be improved:
- X-axis labels in Figs. 7(a) and 7(b) should be made uniform; the Y-scale is too condensed. 
- Fig. 6: y-axis has no tick marks and labels.

4. Other remarks:
- Introduction: sentences like "photon-to-electron transformation within them" should be rephrased
- QL defined in Section 3-4, first use in Section 3-2;
- In part, the English should be revised

Round 2

Reviewer 2 Report

Authors revised according to the reviewer's comment. So it can be accepted as it is.